# TEXTURAL OR TEXTUAL: HOW VISUAL MODELS UNDERSTAND TEXTS IN IMAGES

## ABSTRACT

It is widely assumed that typographic attacks succeed because multimodal pre-trained visual models can recognize the semantics of text within images, allowing text to interfere with image understanding. However, the assumption that these models truly comprehend textual semantics remains unclear and underexplored. We investigate how the CLIP encoder represents textual semantics and identify the mechanisms through which text disrupts visual semantic understanding. To facilitate this analysis, we propose a novel ToT (Texture or Textual) dataset, which includes a paronyms-synonyms pair of subsets that disentangles orthographic forms (i.e., the visual shape of words) from their semantics. Using Intrinsic Dimension (ID) to assess layer-wise representation complexity, we examine whether the representations are built on texture or textual information under typographic manipulations. Contrary to the common belief that semantics are progressively built across layers, we find that texture and semantics compete in the early layers. In the later layers, semantic accuracy improves mainly through texture learning that aids orthographic recognition, while a semantically driven representation emerges only in the final block.

## 1 INTRODUCTION

While visual models trained with vision-language supervision demonstrate the ability to interpret text within images, a fundamental question remains: do these models capture the semantic meaning of the text, or do they simply recognize it as a visual pattern? This distinction is critical, as textual elements, despite their inherently different texture properties compared to visual objects, are often encoded in a similar manner. Such encoding raises the possibility that these models achieve only a superficial alignment of textures rather than a deeper cross-modal understanding. Furthermore, as neural networks abstract textual information through successive layers, the point at which textual input begins to influence the semantic interpretation of an image remains uncertain. These challenges form the core motivation for our study, which aims to disentangle textual and textural representations to better understand the semantic mechanisms of vision models.

One significant manifestation of these uncertainties is found in typographic attacks (Goh et al., 2021), which expose the vulnerabilities inherent in contemporary vision-language models when interpreting text within images. These attacks involve embedding misleading text into images, resulting in substantial impacts on recognition and classification accuracy. For instance, an image of a dog superimposed with the word "laptop" may be misclassified as an electronic device (Lemesle et al., 2022). The consequences of such misclassifications depend significantly on the characteristics of the text and image. As models like GPT-4 (vision) (Yang et al., 2023) become more advanced, their susceptibility to these attacks raises important security concerns. These typographic manipulations can lead to unintended command executions, akin to 'jailbreaking' the model (Gong et al., 2023; Robey et al., 2023; Wang et al., 2023).

While typographic attacks may not strictly qualify as traditional attacks, they demonstrate how pre-trained models effectively learn multimodal representations. Models trained on diverse image-text datasets implicitly learn correlations between text and its real-world meanings (Cao et al., 2023). For instance, a model might link the image of a 'cat' with the word and concept of a cat, suggesting a unified representation of textual and conceptual semantics. However, this theory requires further empirical validation, and alternative explanations should also be explored.

In this work, we use Intrinsic Dimension (ID) as a measure of the complexity of data representation, capturing the required degrees of freedom for accurate encoding. ID is particularly useful in understanding how subtle image perturbations, including those introduced by typographic attacks, affect visual models (Amsaleg et al., 2017; Ma et al., 2018). Unlike pixel-level perturbations, typographic attacks involve semantic changes that impact how models represent text in images. Our study extends the use of ID to examine how these semantic variations influence model representations across different layers.

We introduce the ToT (Textural or Textual) typographic attack dataset, comprising words that are consistent, irrelevant, or nonsensical in relation to image content. This dataset enables an examination of how a vision-language pre-trained model processes these varying types of texts. Additionally, We create a subset of 10 paronyms-synonyms pairs to investigate how representations progressively form to distinguish visual similarities from semantics. Our findings reveal a non-linear pattern in representation; while texture representation evolves gradually in the earlier layers, significant shifts in semantic understanding occur only in the final network block. Specifically, the main contributions of this work are as follows:

- We present a detailed analysis of typographic attacks on visual models, examining their processing of textual content within images. Through intrinsic dimension (ID), we find that texture and semantic representations share significant features in most layers. Initially, Texture and textual representations compete in the early layers. As the layers progress, the complexity of the representation increases, and texture representation grows rapidly. While the complexity begins to decrease, semantic understanding improves but relies on texture learning for orthographic recognition. Notably, a semantically driven representation emerges only in the final block.

- Building on these observations, we defend against typographic attacks by simply fine-tuning only the final block of the model to better distinguish between textural and textual representations. Experimental results show that our strategy effectively balances the performance between the original image and the typographic classification, achieving significant improvements across diverse defense scenarios.

## 2 RELATED WORK

### 2.1 TYPOGRAPHIC ATTACK

CLIP (Radford et al., 2021) is known for its ability to joint understanding of language and vision. Due to its large amount and spin of training images, many of which incorporate both visual and textual features, it can read visually presented text, or scene-text (Materzyńska et al., 2022; Cao et al., 2023). A notable aspect of CLIP is its tendency, in certain instances, to rely predominantly on text for image classification. This reliance can lead to what's termed a typographic attack (Goh et al., 2021), where misclassification occurs due to overemphasis on text.

In response to such vulnerabilities, various defense strategies have been explored. Materzyńska et al. (2022) implement a linear transformation to bifurcate the model into two distinct streams: one dedicated to visual information and the other to textual data. Azuma & Matsui (2023) introduce the Dense-Prefix token in conjunction with prompt learning, placing it before class names to significantly enhance accuracy against real-world typographic attack scenarios. PAINT Ilharco et al. (2022) involves a method that interpolates between a model's pre- and post-fine-tuning weights, showing notable success in mitigating typographic attacks. Cao et al. (2023) takes a different route by filtering out dataset samples containing text regions within images, leading to not only improved defense against typographic attacks but also heightened accuracy in other tasks.

### 2.2 DISENTANGLING VISUAL AND TEXTUAL SEMANTICS IN VISION-AND-LANGUAGE MODELS

Large vision-and-language pre-trained models like CLIP (Radford et al., 2021) showcase their efficacy through extensive pre-training on diverse datasets, excelling in tasks such as image classification (Zhang et al., 2022), visual question answering (VQA) (Song et al., 2022), and image captioning (Mokady et al., 2021). The treatment of visually presented text within these models sparks debate in

the field. Some researchers recommend removing the language representation from the visual aspects of the model (Materzyńska et al., 2022; Cao et al., 2023). In contrast, other researchers underscore the indispensable role of language comprehension in tasks like Text-VQA and Text-Captioning (Yang et al., 2021; Kil et al., 2023). They advocate for a harmonious integration of visual and textual information, pointing out that such synergy is crucial for a more holistic understanding of images.

In line with this debate, our research undertakes a series of comparative experiments focusing on CLIP's Vision Transformer (Dosovitskiy et al., 2020). These experiments aim to unravel the intricate dynamics between scene-text recognition and the multi-modal properties inherent in CLIP. Addressing the complexities of multi-modal models, particularly their challenge in differentiating visual elements from textual semantics, our study seeks to fine-tune this delicate balance. We endeavor to enhance the model's capability to discern physical objects from scene text, thereby enriching its understanding and interpretation of both visual and textual components in a unified and coherent manner.

### 2.3 INTRINSIC DIMENSION IN ADVERSARIAL ATTACK

The Intrinsic Dimension (ID) is the minimum number of dimensions required to represent data effectively (Levina & Bickel, 2004). In neural networks, ID is derived from the model's representations, indicating the fewest parameters needed to capture specific features (Amsaleg et al., 2015). Ansuini et al. (2019) demonstrated a correlation between the final layer's ID and the model's accuracy, noting that ID typically follows a hunchback-shaped curve across layers, reflecting the learning process (Ansuini et al., 2019). Moreover, ID is crucial for interpreting learned representations and exploring its relationship with neural network training (Aghajanyan et al., 2020; Pope et al., 2021).

Amsaleg et al. (2017) and Ma et al. (2018) used local ID to assess adversarial robustness, finding that LID increases with noise in adversarial perturbations. This connection emphasizes how ID influences a model's vulnerability. Tulchinskii et al. (2024) further explored ID in textual data, revealing that human-generated texts have an average ID of 7 to 9, while AI-generated texts often fall below 1.5. This distinction enables classifiers to effectively differentiate between human and AI-generated content.

## 3 METHOD

### 3.1 ToT (TEXTURAL OR TEXTUAL) DATASETS

#### 3.1.1 SUBSET 1: SEMANTIC CONFUSION

We propose the ToT (Textural or Textual) dataset, derived from ImageNet-1k, which features 100 categories of common objects overlaid with texts of varying semantics. The dataset contains 50,000 images, with 500 randomly selected images per category. These categories represent frequently encountered real-world objects with short, distinct names and minimal semantic overlap, making the dataset highly relevant for studying typographic attack scenarios in practical contexts. Figure 1 illustrates the three types of textual modifications applied to the images to generate a diverse set of compositions.

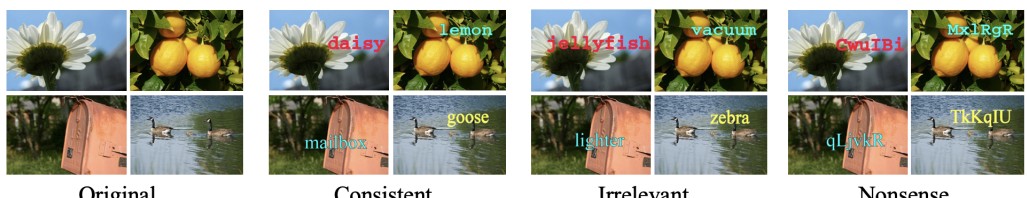

| Original | Consistent | Irrelevant | Nonsense |

Figure 1: Example images from our ToT (Textural or Textual) dataset, demonstrating consistent, irrelevant, and nonsensical text superimpositions.

**Original.** The unmodified ImageNet-1k images serve as a control group, establishing a baseline for comparative analysis. **Consistent.** Texts corresponding to the image's category from the ToT dataset are superimposed, allowing the evaluation of the model's ability to represent information with

consistent semantics across different modalities. **Irrelevant.** Images are paired with unrelated text from the ToT dataset. For example, an image of a 'candle' might be overlaid with the text 'tiger.' This subset contrasts with the "Consistent" subset, providing a dataset where text disrupts the semantic understanding of the image. **Nonsense.** Images are overlaid with nonsensical strings, formed from random combinations of letters averaging six characters in length, similar in structure to the dataset's category names. This subset allows for the evaluation of the model's ability to distinguish between meaningful words and random characters. For instance, the string 'MxlRgR' might be superimposed on an image.

### 3.1.2 SUBSET 2: VISUAL VS. SEMANTIC CONFUSION

Since the form of a word is often intrinsically linked to its meaning, variations in word structure typically lead to words with distinct semantic differences. This suggests that neural networks may distinguish words based solely on superficial textural features, leading to what appears to be semantic-level comprehension. To explore this hypothesis, we design a subset of 10 word pairs specifically aimed at disentangling the relationship between word form and meaning.

This subset explores how models differentiate between words that are visually similar but semantically distinct, as well as those that share semantic meaning but have different visual forms. Each of the 10 word pairs consists of a base word (selected from the ToT dataset) and two related words: the **Paronyms Pair**, which refers to words that are visually similar but differ in meaning, and the **Synonyms Pair**, which refers to words that have similar meanings but distinct spellings. All words are real-world entities and are commonly used. For example, as shown in Figure 2, 'goose' is paired with 'moose' as its Paronyms Pair and 'gander' as its Synonyms Pair. This dataset enables a detailed analysis of how models process both visual and semantic similarities in language.

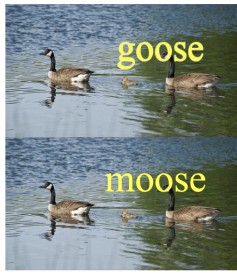 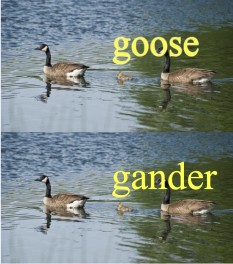

Paronyms Pair          Synonyms Pair

Figure 2: Examples of paronyms and synonyms pairs in the ToT typographic datasets.

### 3.2 ESTIMATING THE INTRINSIC DIMENSIONS OF TOT DATASETS

For images containing varying levels of semantic complexity, we estimate the Intrinsic Dimension (ID) of their representations layer by layer. This process involves using the ID's magnitude as a metric to evaluate how specific layers of the model articulate the textual semantics embedded within the images.

To clarify the scope of our study, we focus on ViT-based vision models, particularly the CLIP ViT-B/16 (cli), due to their dominance in multimodal pretraining and widespread use in current large vision-language models (LVLMs). This model employs a Vision Transformer architecture consisting of 12 blocks, and our analysis concentrates on the output representations from each of these blocks. Within each block, we evaluate three sequential layers:

---

**Algorithm 1** Intrinsic Dimension Estimation Across Network Layers

---

**Require:** $n$: Number of images
  $\Lambda$: Number of network layers
  $model(\cdot, \lambda)$: Image representation from layer $\lambda$
**Ensure:** $ID$: Estimated intrinsic dimensions per layer
  $S \leftarrow$ select $n$ random images
  **for** $\lambda = 1$ to $\Lambda$ **do**
    $F[\lambda] \leftarrow model(S, \lambda)$
    **for** each $i$ in $n$ **do**
      $N_1, N_2 \leftarrow$ nearest neighbors of $F[\lambda][i]$
      $d_1, d_2 \leftarrow$ distances to $N_1, N_2$
      $R[i] \leftarrow d_1/d_2$
    **end for**
    $ID[\lambda] \leftarrow$ linear regression on $R$
  **end for**
  **return** $ID$

---

**Attn**, the output after the attention linear transformation; **c_fc**, a linear layer that projects input features from 768 dimensions to 3072 dimensions; and **c_proj**, which projects features back from 3072 dimensions to 768 dimensions.

The TwoNN algorithm (Facco et al., 2017) quantifies the intrinsic dimension (ID) of visual representations within the dataset. Algorithm 1 illustrates the procedure applied to layers in a pre-trained model. The function model$(S, \lambda)$ extracts the representation of layer $\lambda$ for the image set $S$, while $ID[\lambda]$ holds the estimated IDs for each layer.

To estimate the intrinsic dimension (ID) using the TwoNN method, we first compare the distances to the nearest neighbors, denoted as $d_1$ and $d_2$, and compute the ratio $R[i] = \frac{d_1}{d_2}$ for each sample. As the intrinsic dimension increases, the ratio $R$ typically decreases, which is consistent with a Pareto distribution represented by $Pa(d + 1)$. The relationship is described by the likelihood function:

$$P(\mathbf{R}|d) = d^N \prod_{i=1}^{N} R[i]^{-d-1}$$

$P(\mathbf{R}|d)$ represents the likelihood of observing the vector $\mathbf{R}$ given a particular intrinsic dimension $d$. Linear regression is then applied to maximize this likelihood function, allowing for the estimation of the intrinsic dimension that best captures the local structure of the data.

## 4 ANALYSIS OF TEXTUAL AND TEXTURAL REPRESENTATIONS ACROSS LAYERS

### 4.1 LAYER SENSITIVITY TO TYPOGRAPHIC ATTACKS

We begin by examining how different layers of visual models respond to semantic variations, clustering image representations that include various textual overlays. Since t-SNE provides a limited two-dimensional view of these representations, we then estimate the intrinsic dimensions of each layer, enabling us to assess whether the intermediate layers capture semantic distinctions in higher-dimensional spaces.

**Representation Clustering.** We sample image representations from different network depths and visualize them with t-SNE (Van der Maaten & Hinton, 2008), as shown in Figure 3. In the initial four samplings, the representations appear to cluster into two groups, seemingly influenced by the image background content. In contrast, the final sampling shows eight clusters that align with a combination of the image and text semantics. These patterns suggest the possibility that multi-modal visual models may process text as a textural feature in the earlier layers, with a shift toward capturing semantic information in later layers. This hypothesis is further examined in the next section by estimating the intrinsic dimensions (ID) of representations across different depths.

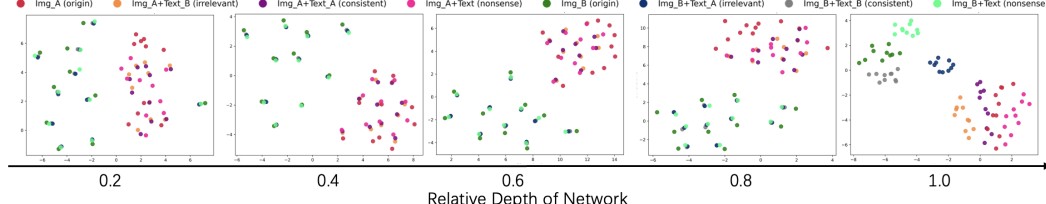

Figure 3: t-SNE visualization of representations with varied semantics, sampled at different depths. Only the final depth distinctly separates all representations, independent of solely image or text semantics.

**Intrinsic Dimensionality Estimation.** We randomly sampled 2,000 images from each subset in the ToT dataset to estimate the Intrinsic Dimension (ID) of representations of the CLIP visual model. The results are illustrated in Figure 4. A swell-shrink pattern is observed across the network layers, where representation complexity first increases and then decreases. This pattern, previously identified in CNN visual models (Ansuini et al., 2019; Muratore et al., 2022), also appears in Transformer-based models, aligning with the information bottleneck theory Shwartz-Ziv & Tishby (2017), which describes an initial fitting phase followed by a compression phase.

Despite the fluctuating ID values, their ratios to the original image's ID remain stable, with typography generally increasing representational complexity by 1.2 to 1.3 across most layers.

However, in the final block, nonsensical and irrelevant subsets show significantly higher IDs than the original, while consistent images exhibit a notable decrease. This discrepancy, particularly pronounced in the last layer closest to the classification layer, suggests that the final block has a significant impact on the semantic representation of the entire image.

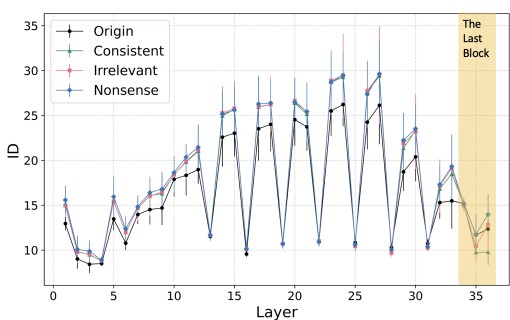

Figure 4: The Intrinsic Dimension (ID) variations across layers for the ToT typographic datasets.

Overall, typography increases the complexity of representations in the intermediate layers, regardless of the semantic relationship between the image and text. However, in the final block, text overlay primarily influences the semantic aspect of representations. Notably, when the text closely relates to the image content, it appears to reduce representational complexity, as indicated by lower ID values in this layer.

## 4.2 DISENTANGLING TEXTUAL AND TEXTURAL REPRESENTATIONS ACROSS LAYERS

To further investigate the findings in Section 4, we design two experiments to disentangle text orthography from its meaning and analyze their effects on image representation. Semantic Constancy examines how varying font sizes influence texture-level representations while preserving semantic consistency. Linear Probe uses Paronyms-Synonyms pairs to evaluate the model's progressive disentanglement and formation of textual and textural representations across layers. Though differing in implementation, both experiments investigate the disentanglement of textual and textural representations.

**Semantic Constancy with Varying Font Size.** As illustrated in Figure 8, images containing text of varying sizes are often perceived as semantically identical, even though their visual appearances differ. This observation prompts an investigation into how a visual model interprets this complexity. We compare the performance of a multimodal CLIP model with a pure visual ViT/B-16 model across different text sizes and semantic contexts. To ensure a fair comparison, a consistent network structure is used for both models. The ViT/B-16 model (Dosovitskiy et al., 2022) pre-trained on the ImageNet-1k dataset (Russakovsky, 2015), serves as the baseline for pure vision models. Both models are tested for accuracy on the ToT typographic dataset, with the results shown in Table 1.

Table 1: Accuracy (%) of the visual model from CLIP and ViT/B-16 pretrain on the ToT dataset with different semantics and font sizes, with numbers representing font sizes.

|      | Orig | Cons_80 | Nons_80 | Irr_20 | Irr_40 | Irr_60 | Irr_80 | Irr_100 | Irr_120 |
|------|------|---------|---------|--------|--------|--------|--------|---------|---------|
| CLIP | 86.6 | **98.4** | 80.4 | 78.8 | 60.7 | 49.2 | 42.9 | 40.5 | 38.9 |
| ViT  | **91.1** | 86.8 | 86.3 | 91.0 | 89.9 | 88.9 | 87.1 | 85.9 | 84.2 |

The multimodal CLIP model demonstrates considerable sensitivity to both word semantics and visual form. For example, its accuracy achieves 98.4% in the 'Cons_80' condition with relevant text, while it declines to 80.4% in the 'Nons_80' condition with irrelevant text, even though the font size remains consistent. This observation suggests that visual models trained with vision-language supervision are influenced by both the semantic relevance of the text and its textural complexity, with an increase in irrelevant text size leading to further decreases in accuracy.

In contrast, the purely visual ViT model primarily perceives text as a visual disturbance. Its accuracy decreases from 91.1% to 84.2% as font size increases, irrespective of the semantics involved. This implies that the performance of pure visual models is mainly affected by text size or texture, rather than the meaning of the text itself.

Overall, these findings indicate that multimodal models like CLIP are affected by both semantic and visual aspects of text, while purely visual models such as ViT/B-16 are largely influenced by visual complexity.

**Linear Probe on Paronyms-Synonyms Pairs.**
We apply a linear binary classifier probe to the final outputs (ln_2 layer) of all 12 Residual Attention Blocks for both Synonyms and Paronyms pairs. For each pair in subset 2 of the ToT dataset, we use 320 image samples for training and 80 for testing. Following the approach used in CLIP's linear probe experiments, we employ logistic regression as the classifier.

As shown in Figure 5, each lighter-colored line represents an orthographically similar pair (pink) or a semantically similar pair (orange), the darker lines indicate the average accuracy of the corresponding 10 pairs. It is evident that all layers achieve higher accuracy when classifying based on orthographic similarity. However, the layers with the steepest slopes for these curves show a distinct pattern: the significant improve-

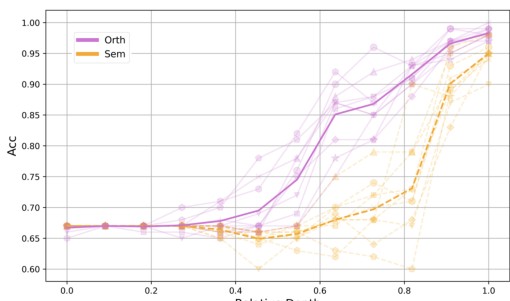

Figure 5: Classifications accuracy of each layer for orthographically similar (pink) and semantically similar (orange) pairs. Lighter lines represent individual pairs, darker lines show the average accuracy of 10 pairs.

ment for texture features occurs primarily in the middle layers, whereas the notable enhancement for textual features is concentrated in the layers closer to the output.

To fully interpret how visual models develop textual representations across layers, we compare the intrinsic dimensions (ID) of Figure 4 with the linear probe results. Our analysis reveals two phases: an initial increase in representational complexity followed by compression. These findings align with the information bottleneck theory Saxe et al. (2019), which describes deep networks undergoing fitting and compression. A closer examination of each layer uncovers the following stages:

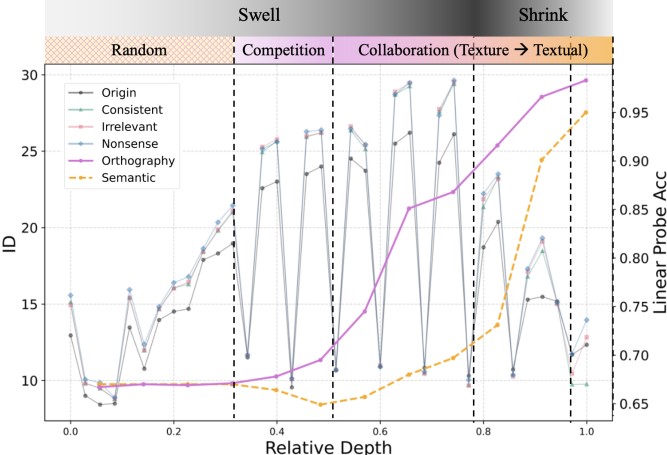

Figure 6: Combination of ID and linear probe results illustrating the progression of visual representations of text across several stages. The semantic accuracy demonstrates a rapid enhancement only as the overall representation complexity decreases.

**Random Initialization.** In this phase, representational complexity gradually increases but contributes little to orthographic or semantic understanding. Classification accuracy remains at the chance level, indicating the model is still in its initial learning stage. **Texture Maximization.** Representational complexity increases significantly, enhancing texture recognition. The accuracy of orthographic classification improves rapidly as the model captures visual features. However, semantic understanding initially competes with texture representation and then rises slowly, yet it remains limited. **Compression and Semantic Integration.** As the model compresses its representations, ID drops

sharply, leading to rapid gains in semantic understanding. By the final block, the model effectively integrates textual and visual features within the semantic space.

In summary, the process can be divided into four phases: initialization, competition, texture-dominated collaboration, and semantic-dominated collaboration. Our key findings are:

**Shared Features Between Textual and Textural Representations.** In most layers, the features that contribute to both texture and semantic representations are identical and shared, however, the development of semantic representations lags behind that of texture.

**Misleading Semantic Understanding via Texture.** In the early stages of representation complexity reduction (the shrink stage), a high level of semantic understanding is achieved; however, this understanding is primarily based on texture features, akin to the recognition of word orthography.

**Semantic Understanding in the Final Block.** A clear distinction in intrinsic dimension (ID) between semantics emerges only in the final block, indicating that semantically focused understanding is achieved after processing textural information in previous layers.

### 4.3 Correlation Between ID and Accuracy

Table 2 shows the relationship between the ID of the last fully connected layer (ID_Last), the maximum ID (ID_Max), and classification accuracy for ViT and CLIP models. The Spearman correlation coefficient Spearman (1987) is used to measure the correlation between accuracy and ID values.

The correlations for ViT are $\rho(\text{ID\_Last}, \text{Acc}) = -0.98$ and $\rho(\text{ID\_Max}, \text{Acc}) = -0.63$, while for CLIP, they are $\rho(\text{ID\_Last}, \text{Acc}) = -0.13$ and $\rho(\text{ID\_Max}, \text{Acc}) = -0.73$. Overall, an inverse correlation is observed, suggesting that lower ID values correspond to higher classification accuracy. However, this trend is not consistent for the last layer ID, deviating from patterns typically found in standard image classification tasks Ansuini et al. (2019).

Table 2: Correlation between classification accuracy and ID values for the last layer and maximum ID across layers, differentiated by typography type and size. The models compared are pre-trained via multimodal (CLIP) and pure vision (ViT) models.

| Model | | Orig | Cons_80 | Nons_80 | Irr_20 | Irr_40 | Irr_60 | Irr_80 | Irr_100 | Irr_120 |
|-------|--------|------|---------|---------|--------|--------|--------|--------|---------|---------|
| ViT | Acc. | 91.1 | 86.8 | 86.3 | 91.0 | 89.9 | 88.9 | 87.1 | 85.9 | 84.2 |
| | ID_Last | 6.8 | 7.8 | 8.0 | 7.0 | 7.2 | 7.6 | 7.8 | 8.1 | 8.3 |
| | ID_Max | 89.9 | 92.4 | 95.4 | 95.6 | 89.1 | 95.2 | 100.0 | 99.1 | 102.5 |
| CLIP | Acc. | 86.6 | 98.4 | 80.4 | 78.8 | 60.7 | 49.2 | 42.9 | 40.5 | 38.9 |
| | ID_Last | 12.4 | 9.8 | 14.0 | 14.8 | 14.3 | 13.0 | 12.7 | 12.5 | 12.6 |
| | ID_Max | 26.2 | 29.4 | 29.6 | 29.2 | 29.5 | 29.4 | 29.5 | 29.8 | 29.8 |

For the ViT model, there is a clear correlation between accuracy and text size: as text size increases, accuracy decreases, which aligns with changes in ID_last values. This observation supports findings from Ansuini et al. (2019), where text size plays a significant textural role in pure vision models.

In contrast, the CLIP model shows that text semantics significantly impact accuracy, even when size is controlled. The relationship between accuracy and ID metrics is more complex here; while no clear correlation exists with ID_last for semantically irrelevant texts, there is a strong inverse correlation between accuracy and ID_max as text size increases. This suggests that ID_max captures textural complexity, whereas ID_last reflects both textural and textual features. CLIP's representation of text involves a complex interaction between these elements, with semantics heavily influencing accuracy, yet no single layer fully captures this correlation.

## 5 Defense Against Typographic Attacks through Fine-Tuning

Based on the observations in Section 4, different layers of the visual model encode text in distinct ways, as indicated by the intrinsic dimension and linear probe results. These layers can be grouped into three categories, each focusing on different aspects of visual or semantic representation. Depending on the defense requirements, such as whether understanding the text's meaning is necessary, it is

possible to selectively fine-tune specific layers for defense. To verify this, we design three typographic attack tasks of varying difficulty: **Easy**: Recognizing image content while ignoring text, similar to the setup in most typography attack work (Materzyńska et al., 2022; Ilharco et al., 2022; Azuma & Matsui, 2023). **Medium**: Detecting the presence of text without understanding its meaning. **Hard**: Distinguishing the semantics of both text and image.

The progression from easy to hard illustrates the increasing complexity of semantic understanding required at each stage. Ideally, fine-tuning only the swell blocks should not effectively defend against any level of attack. In contrast, fine-tuning the shrink and last in shrink blocks should provide varying levels of defense based on semantic comprehension. For example, medium difficulty may only require recognition of word orthography, necessitating adjustments to the shrink blocks, while the hard level requires understanding specific meanings, thereby requiring fine-tuning of the last block for effective defense.

Figure 7: Examples of image-text pairs of easy to hard defense level.

All of our experiments are conducted on a GeForce RTX 3090 GPU. We use a batch size of 512 and a learning rate of $1 \times 10^{-4}$, with a weight decay of 0.2. The Adam optimizer is employed for training.

## 5.1 BLOCK-SPECIFIC FINE-TUNING FOR TEXTUAL AND TEXTURAL CONTROL

We divide the CLIP encoder into three sections: Swell, Shrink-Last, and Last, as described in Section 4. We fine-tune each section on the hard-level task and evaluate their performance across easy, medium, and hard tasks. The results are summarized in Table 3.

Fine-tuning the Swell block alone yields suboptimal performance across all difficulty levels, particularly in tasks requiring semantic understanding. Fine-tuning the Last block proves most effective, particularly in handling higher complexity tasks like Hard-Nons (81.7%) and maintaining high Orig performance (84.7%).

The Shrink strategy also performs well, especially in tasks requiring nuanced text-image understanding, with strong results in Medium and Hard categories (70.8% in Hard-Irr). However, fine-tuning the Shrink-Last module provides a balanced performance, almost matching Last in the most difficult tasks while still lagging slightly in simpler cases like Orig (83.5%). This suggests that while Shrink-Last captures some mid-layer texture refinement, it is not as adept at final-stage semantic comprehension as Last alone.

Table 3: Performance comparison when fine-tuning different partial of the CLIP visual encoder.

| Fine-tuned Blocks | Orig | Easy | | | Medium | | | Hard | | |
|---|---|---|---|---|---|---|---|---|---|---|
| | | Cons | Irr | Nons | Cons | Irr | Nons | Cons | Irr | Nons |
| CLIP w/o ft | 82.3 | 97.3 | 50.6 | 73.7 | 94.5 | 65.9 | 77.4 | 14.5 | 59.9 | 77.2 |
| Swell | 62.8 | 85.8 | 39.1 | 51.2 | 79.0 | 52.1 | 56.3 | 6.6 | 29.7 | 56.2 |
| Shrink | 82.6 | 98.6 | 43.0 | **77.0** | 97.8 | 68.9 | 81.3 | 32.6 | **70.8** | 81.1 |
| Shrink - Last | 83.5 | **98.7** | 32.0 | 76.6 | **98.1** | 61.2 | **81.4** | **36.0** | 68.9 | **81.7** |
| Last (Ours) | **84.7** | 98.2 | **60.0** | 76.8 | 96.7 | **74.5** | 81.0 | 35.0 | 69.5 | **81.7** |

## 5.2 DEFENSE VIA FINE-TUNING THE LAST BLOCK

### 5.2.1 DEFENSE WITH IGNORING TYPOGRAPHY

**Setup.** Following common implementations, we fine-tune the model using subsets of original and irrelevant images at the easy level, in a task that is similar to ignoring the text content. To evaluate our method in real-world scenarios, we utilize publicly available typographic attack datasets, including Disentangle (Materzyńska et al., 2022), PAINT (Ilharco et al.,

2022), and Prefix (Azuma & Matsui, 2023), which contain images with handwritten texts on notepads. We perform cross-testing on each dataset using the methods from these studies.

**Results.** Table 4 compares the performance of SOTA defense methods. Notably, the Prefix method fine-tunes only the language model, while other approaches involve retraining both the vision and language models. Our method outperforms the comparison methods across all datasets, although it is slightly surpassed by the Disentangle method on the Disentangle dataset.

Table 4: Comparison to SOTA defense methods on handwritten typographic datasets.

| Method | Disentangle | PAINT | Prefix | Avg. |
|---|---|---|---|---|
| CLIP | 43.3 | 50.0 | 47.2 | 46.8 |
| Disentangle | **77.8** | 55.5 | 57.6 | 63.6 |
| PAINT | 53.2 | 58.2 | 53.6 | 55.0 |
| Prefix | 71.9 | 63.6 | 58.0 | 64.5 |
| Ours | 73.3 | **68.2** | **67.0** | **69.5** |

### 5.2.2 DEFENSE WITH PRESERVING THE TYPOGRAPHY SEMANTICS

Table 5 presents the evaluation results for medium and hard levels of defense, which require the recognition of the absence of words and specific semantics, respectively. Our method outperforms other models across all difficulty levels. The Prefix and Disentangle methods, trained on datasets similar to those used for easy-level tasks, reveal limitations in recognizing character forms and semantics, as demonstrated by their performance in the hard-level results. In contrast, our model exhibits superior comprehension across various difficulty levels, particularly when the image-text relationship is semantically consistent.

Table 5: Accuracy comparison with CLIP, Prefix (Azuma & Matsui, 2023), Disentangle (Materzyńska et al., 2022) methods on various levels of defense.

| Method | Orig | Medium | | | Hard | | |
|---|---|---|---|---|---|---|---|
| | | Cons | Irr | Nons | Cons | Irr | Nons |
| CLIP | 82.3 | 94.5 | 66.0 | 77.4 | 14.5 | 59.9 | 77.2 |
| Prefix | 82.0 | 91.4 | 69.9 | 76.0 | 10.6 | 27.1 | 75.5 |
| Disentangle | 79.9 | 85.0 | 72.0 | 75.2 | 12.9 | 13.8 | 75.3 |
| Ours_Med | 83.5 | 92.2 | **82.4** | **82.9** | 8.1 | 22.0 | **82.4** |
| Ours_Hard | **84.7** | **96.7** | 74.5 | 81.0 | **35.0** | **69.5** | 81.7 |

Training on datasets with higher difficulty levels presents challenges in balancing 'Cons' and 'Irr' image-text pairings in the medium scenarios. However, in hard scenarios, where understanding both textual and visual semantics is essential, performance can be improved simultaneously. With the appropriate training data, our method effectively fine-tunes models to comprehend both textual and visual semantics.

Another advantage of our approach is its ability to balance adversarial tasks with the original task. As shown in the 'orig' column of Table 5, our methods outperform all other models, despite primarily being trained on typographic samples. Notably, the 'Ours_Hard' model demonstrates improved 'orig' accuracy, even when typographic semantics potentially conflict with original image classification.

## 6 CONCLUSION

We explore how visual models process textual semantics in the context of typographic attacks. By introducing the ToT dataset and applying Intrinsic Dimension (ID) analysis, we reveal that early layers of visual models primarily rely on texture features rather than true semantic understanding. Only in the final block do models construct a semantically focused understanding after significant compression of textual information. Furthermore, we demonstrate an effective defense strategy by fine-tuning the final block, which enhances the model's ability to distinguish between textural and textual elements. This approach significantly improves performance across various defense scenarios, offering a practical solution to typographic attacks.

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

## A APPENDIX

### A.1 DETAILS OF THE TOT DATASETS

To create the Textural or Textual (ToT) dataset, we follow the approach of PAINT and Prefix. We resize the images to 224 pixels in the shorter dimension using bicubic interpolation and crop a 224x224 pixel area from the center, consistent with standard CLIP resizing and cropping techniques. The text is randomly overlaid at arbitrary positions on the images.

**Font.** We randomly select from Roman, Courier, and Times fonts and utilize eight colors: black, blue, cyan, green, magenta, red, white, and yellow. The text is outlined with a 1-point shadow in a contrasting color.

**Font sizes.** We use 80 points to generate images for the Consistent, Irrelevant, and Nonsense categories. Additionally, to further investigate the impact of font size on identification (ID), Irrelevant images are created in font sizes ranging from 20 to 120 points. The examples are shown in Figure 8.

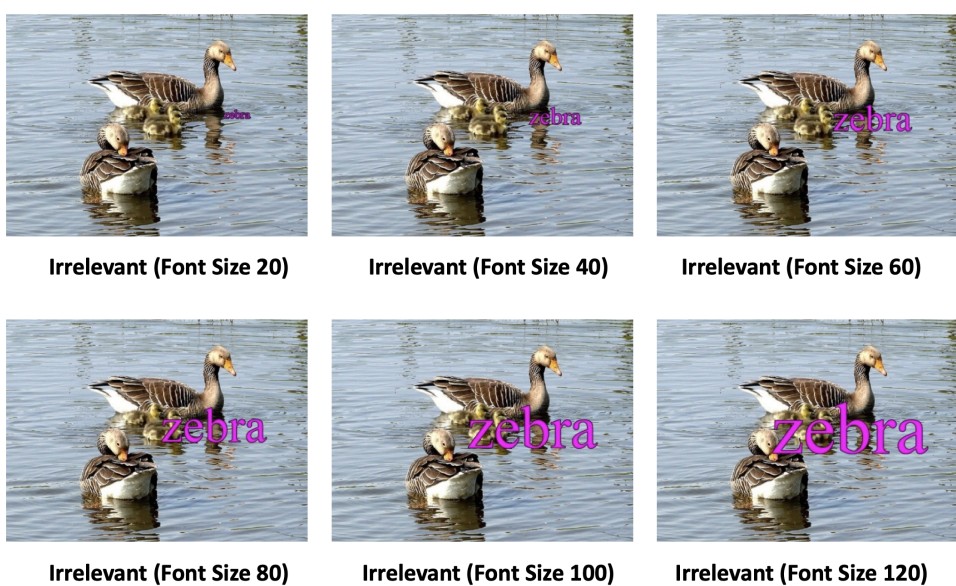

Figure 8: Examples of typography with different sizes.

**Category of Subset 1.** The 100 categories of the ToT datasets are peacock, goose, koala, jellyfish, snail, flamingo, sea lion, Chihuahua, tabby cat, lion, tiger, bee, dragonfly, zebra, pig, llama, panda, backpack, barrel, basketball, bikini, bottlecap, bow, broom, bucket, buckle, candle, cannon, cardigan, carton, coffee mug, coffeepot, crib, envelope, fountain, iPod, iron, jean, ladle, laptop, lighter, lipstick, lotion, mailbox, mask, microwave, mitten, mouse, nail, necklace, paddle, pajama, perfume, pillow, plastic bag, printer, projector, purse, radio, refrigerator, ruler, shovel, sock, stove, suit, sunglass, swing, switch, table lamp, teapot, television, toaster, tray, tub, umbrella, vacuum, vase, violin, wallet, whistle, ice cream, bagel, hotdog, cucumber, mushroom, Granny Smith, strawberry, orange, lemon, banana, hay, dough, pizza, potpie, red wine, espresso, cup, volcano, daisy, and corn.

**Category of Subset 2.** The subset includes the following paronyms and synonyms pairs: Goose (n01855672): Moose, Gander; Bee (n02206856): Beef, Wasp; Pig (n02395406): Fig, Hog; Fountain (n03388043): Mountain, Spring; Mitten (n03775071): Kitten, Glove; Nail (n03804744): Mail, Spike; Hay (n07802026): Ray, Straw; Espresso (n07920052): Express, Coffee; Lemon (n07749582): Demon, Lime.

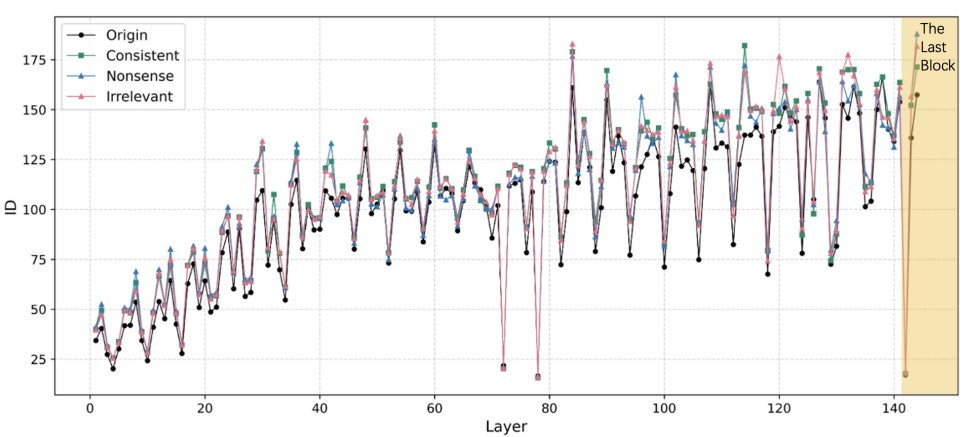

Figure 9: ID Variations of the ShareGPT-4v's Visual Encoder.

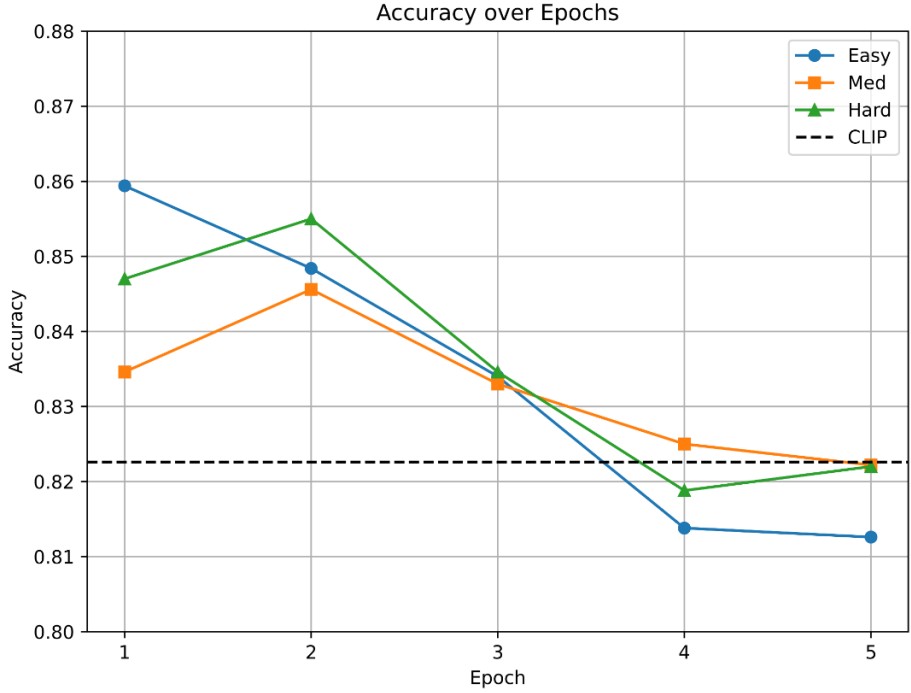

Figure 10: Original Task Accuracy Over 5 Epochs of Our Fine-Tuning.

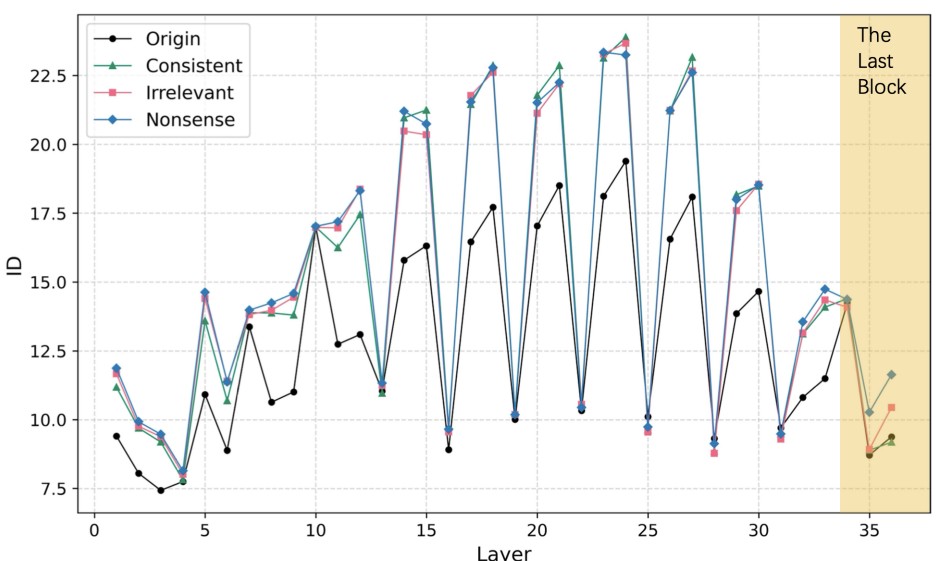

Figure 11: ID Variations with Different Typography Overlaid on the Caltech101 Dataset.

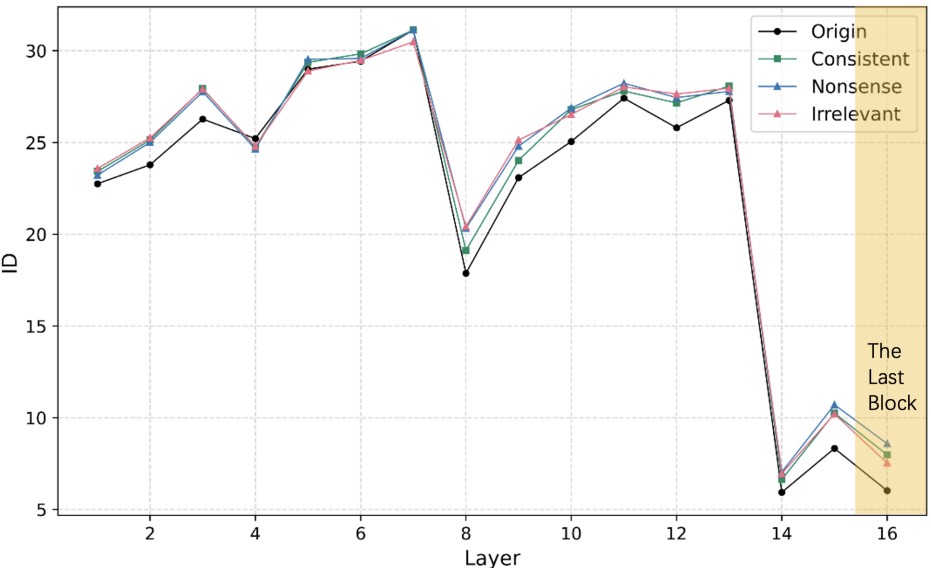

Figure 12: ID Variations of the ResNet-50×4 Encoder in CLIP.

