# OpenReview forum: "Textural or Textual: How Visual Models Understand Texts in Images"
_ICLR.cc/2025/Conference — Submitted to ICLR 2025_

### Official Review · Reviewer_zZ4k · 2024-10-28

**Soundness:** 3
**Presentation:** 3
**Contribution:** 3
**Rating:** 6
**Confidence:** 2

**Summary:**

This paper challenges the assumption that multimodal pretrained visual models, like CLIP, effectively comprehend textual semantics within images. It investigates how the CLIP encoder represents textual semantics and how text disrupts visual understanding. To facilitate this, the authors introduce a new dataset, ToT (Texture or Textual), which separates orthographic forms from their semantics. Their analysis reveals that texture and semantics compete in early layers, and while semantic accuracy improves in later layers, this is largely due to texture learning. Genuine semantic representation is only constructed in the final layer of the model.

**Strengths:**

1. This paper conducted a more detailed experimental analysis, and the experimental results reveal the layers of visual models mainly depend on texture features instead of authentic semantic understanding. Genuine semantic representation is constructed only in the final block, following substantial compression of the textural information.
2. The paper finetunes the last block based on its findings, resulting in overall better performance in defending against typographic attacks compared to other methods.

**Weaknesses:**

1. The authors do not intuitively explain the motivation for using the Intrinsic Dimension. Although it is introduced in the related work section and Section 3.2, the authors do not emphasize what phenomenon this metric intends to reveal in this paper's context. It makes the experimental results not easily understood in Figure 4.
2. The analysis of the experimental results requires significant effort to understand. In the section on Intrinsic Dimensionality Estimation, lines 268 to 272, the authors discuss the swell-shrink pattern. However, this is not related to the conclusions of this section, which may lead to confusion. The experiment's conclusion on Semantic Constancy with Varying Font Sizes (lines 234-236) indicates that multimodal models are influenced by the semantics of the text. However, the authors do not clarify the connection to the disentangling cross-layer textual and textural representations discussed in Section 4.2.
3. The paper should also present the results of fine-tuning using the **Nonsense** type pairs from the dataset to enrich the experiments of defense against typographic attacks. This type of attack is also common in practice.

**Questions:**

See weaknesses.

---

> ### Author Response · Authors · 2024-11-22
>
> We sincerely thank the reviewer for their valuable suggestions. Below, we provide responses that aim to clarify the concerns and enhance the presentation of our work. Key revisions are highlighted in blue in the PDF.
>
> - Q1. Intuitively explain the motivation for using the Intrinsic Dimension.
>
> 	A1. The motivation for using the Intrinsic Dimension (ID) is explained in the Introduction (Lines 54-60, Page 2). In the revised version, we will refine and expand this section to provide a clearer explanation of our rationale. We hope these revisions will enhance the clarity and flow of the paper.
>
> - Q2.1. Discuss the swell-shrink pattern is not related to the conclusions of this section, which may lead to confusion.
>
> 	A2.1. The swell-shrink pattern is included to provide a comprehensive overview of the ID variation trend, highlighting the transition to the shrink phase. This phase is critical as it marks a reduction in representation complexity and the emergence of semantic representations, which directly relates to our core findings.
>
> 	While the swell-shrink pattern is not the main focus of this section, it plays an important role in introducing the discussion on the emergence of semantic representations, which is central to our analysis. In the revision, we will streamline the explanation of the swell-shrink pattern (Line 266-269 in Section 4.1) to focus on its relevance as a conceptual lead-in to our main conclusions.
>
>
> - Q2.2. The experiment's conclusion on Semantic Constancy with Varying Font Sizes (lines 234-236) indicates that multimodal models are influenced by the semantics of the text. However, the authors do not clarify the connection to the disentangling cross-layer textual and textural representations discussed in Section 4.2.
>
> 	A2.2. Thank you for your question. It prompts us to consider whether the connection between the two experiments in Section 4.2 is clearly conveyed.
>
> 	Section 4.2 aims to disentangle textual and textural representations in multimodal models through two complementary experiments. The first experiment (Semantic Constancy with Varying Font Sizes) uses a simpler manipulation—changing font size—to test whether semantics remain stable despite subtle variations in appearance. The second experiment (Orthography-Semantic Pairs) employs a more explicit manipulation to disentangle visual and semantic components. While these experiments differ in design, they converge on the same conclusion, reinforcing the robustness of our findings.
>
> 	To improve clarity, we revise the introduction of Section 4.2 (Lines 292-297) to better explain the motivations and connections between these experiments. We hope these adjustments will address your concern and enhance the manuscript's coherence.
>
>
> - Q3. The results of fine-tuning using the Nonsense type pairs from the dataset to enrich the experiments of defense against typographic attacks.
>
> 	A3. Thank you for raising the importance of evaluating defenses against nonsense typography, which is a critical challenge in real-world applications.
>
> 	To address this, our experiments already include such cases. Specifically, Table 5 presents results where both the training and testing datasets incorporate examples from all subsets of the ToT dataset, including nonsense typography. The 'Nons' column in Table 5 highlights the defense performance against these attacks, where our method significantly outperforms baseline approaches, demonstrating robust effectiveness.

---

> > ### Comment · Reviewer_zZ4k · 2024-11-24
> >
> > Thanks for the author's response. It effectively addressed my concerns, and I appreciate the clarification. Based on this, I will maintain my original score.

---

### Official Review · Reviewer_41wq · 2024-11-01

**Soundness:** 2
**Presentation:** 2
**Contribution:** 2
**Rating:** 5
**Confidence:** 3

**Summary:**

This paper investigates how the CLIP encoder represents textual semantics and identify the mechanisms through which text disrupts visual semantic understanding. A novel ToT (Texture or Textual) dataset is built on texture or textual information under typographic manipulations. Authors claim to find that texture and semantics compete in the early layers. In the later layers, while semantic accuracy improves, this gain primarily stems from texture learning that aids orthographic recognition. Only in the final block does the visual model construct a genuine semantic representation.  The experiments are thorough.

**Strengths:**

They analyze the representations of semantics and textures in different layers of CLIP more clearly with Intrinsic Dimension.
Also try to construct a reasonable dataset for typographic attack analysis  with extensive experiments.

**Weaknesses:**

The Intrinsic dimension (ID) is interesting, but a more explicit investigation into ID for this task should be well studied.

**Questions:**

A more explicit investigation into ID for this task should be well studied.
As the title is “How visual models understand texts in images”, does this conclusion apply to CNN-based CLIP models or other visual models?
In Table 3 and Table 5, why did the accuracy for Cons show a significant decrease in the Hard case? If the model is only required to identify text within images, how well it perform?
In section 5.2.2, If other methods are trained using the same dataset as yours, how about the performance?

---

> ### Author Response · Authors · 2024-11-22
>
> We sincerely thank the reviewer for their valuable suggestions. Below, we provide responses that aim to clarify the concerns and enhance the presentation of our work. Key revisions are highlighted in blue in the PDF.
>
> - Q1. As the title is “How visual models understand texts in images”, does this conclusion apply to CNN-based CLIP models or other visual models?
>
> 	A1. Thank you for your insightful question. This paper primarily investigates Vision Transformer (ViT) models, which are increasingly dominant in multimodal pre-training and widely used in leading large vision-language models (LVLMs), such as LLaVA, BLIP, LLaMA-Adapter, Phi-3-Vision, TransCore-M, ShareGPT4V, mPLUG-Owl2, and OpenFlamingo. Given their prevalence in contemporary LVLMs, the conclusions drawn from our study are highly relevant and extendable to these models.
>
> 	To clarify the scope of our study, we specify in Section 3.2 that our experiments focus on ViT models. However, to provide a broader context, we also perform a similar intrinsic dimension (ID) analysis on CLIP (ResNet 50*4), with the results shown in Figure 12 (in the appendix). Additionally, we include ShareGPT-4v results in Figure 9 (also in the appendix).
>
> 	These additional analyses and clarifications are intended to show that our conclusions are not only relevant to ViT models but also applicable to a wider range of contemporary vision-language models, helping to address any concerns about the broader applicability of our findings.
>
>
> - Q2. In Table 3 and Table 5, why did the accuracy for Cons show a significant decrease in the Hard case? If the model is only required to identify text within images, how well it perform?
>
> 	A2. The accuracy drop for the "Cons" in the Hard case (Tables 3 and 5) is due to the increased difficulty of this task, where the model must not only detect the presence of text but also understand its exact semantic meaning. This is much more challenging compared to the Medium case (detecting text presence) or the Easy case (ignoring the text's semantics). We elaborate the difference with the example in Figure 7 (in the manuscript). Given the complexity of the Hard task, the accuracy decrease is reasonable.
>
> - Q3. If other methods are trained using the same dataset as yours, how about the performance?
>
> 	A3. Thank you for your question. We understand that you are aiming to separate the impact of the dataset and the method on the results. However, comparing our approach with the other methods may not be entirely fair, as these methods are not specifically designed to address the precise semantic understanding of text in images, as seen in Table 5 of the manuscript.
>
> 	Among the three comparison methods we used (Disentangle, Prefix, and PAINT), each has a different focus. The Prefix method emphasizes language modeling, similar to adversarial word embedding training. PAINT focuses on interpolating the parameters of the entire VLM model. Only the Disentangle method is somewhat comparable to our approach, though it was trained with a setup designed for scenarios like the “irrelevant” (easy) case in our work.
>
> 	To make the comparison as fair as possible, we trained and tested Disentangle on a subset of the data, using only the "original" and "irrelevant" samples, which align with the original Disentangle implementation. As shown in the following Table, Disentangle performs lower on the ToT subset compared to the original CLIP model and its performance on the original Disentangle dataset. This result is understandable, given that the Disentangle dataset is approximately 700 times larger than ToT.
>
> 	Considering the model design and dataset scale, the experiments in Table 4 (in the manuscript) provide the fairest comparison across methods. However, this also highlights the limitations of comparison methods, which focus primarily on image semantics and neglect textual semantics.
>
> | **Model**                | **Orig** | **Irr Easy** | **Irr Med** | **Irr Hard** |
> |--------------------------|----------|--------------|-------------|--------------|
> | CLIP                     | 82.3     | 50.6         | 65.9        | 59.9         |
> | Disentangle on ToT        | 71.4     | 54.4         | 57.8        | 0.6          |
> | Disentangle on Disentangle| 79.9     | 64.3         | 72.0        | 13.8         |

---

> > ### Author Response · Authors · 2024-11-26
> >
> > Dear Reviewer,
> >
> > Thank you for your thoughtful feedback and the time you have devoted to reviewing our work. As the rebuttal period is coming to an end, we hope that the revisions have effectively addressed your concerns. If there’s anything further you’d like to discuss, we’d be happy to engage.

---

### Official Review · Reviewer_3PXM · 2024-11-03

**Soundness:** 2
**Presentation:** 3
**Contribution:** 2
**Rating:** 5
**Confidence:** 4

**Summary:**

The paper discusses the problem of how CLIP confuses  text inside images with visual object itself, and introduces some defenses to typographic attacks.

**Strengths:**

This is an interesting problem, and I also find the dataset proposed interesting.
The defense method is simple and seems to do better than baseline methods used for comparison.

**Weaknesses:**

The paper seems to have parts that are not well connected: the results on the intrinsic dimension (ID) seem disconnected from the defenses and results presented in section 5. It will be better to strengthen the connection to justify why the ID is needed for this paper.

Some parts of the paper would benefit from more clarification. I do not think this is an important weakness as the paper is overall clear. But I include some suggestions later.

**Questions:**

Here there are some suggestions to improve the paper clarity in case they are useful to the authors:

1. I would recommend making some questions softer. For instance, the question “do these models genuinely understand the semantics of the text or are they merely recognizing it as a visual pattern?” is a really difficult question and cannot be answered by the experiments shown in this paper. I do not think the authors need to set such a high bar so early in the paper.

2. Line 68, the sentence “Our findings reveal a non-linear pattern in representation” is repeated twice.

3. Maybe you could rename the orthographic pairs as “Paronyms”: words that are similar in spelling but have different meanings.

4. In the algorithm 1, you first store in R the ratios between the first and second nearest neighbors for all images. Then you compute the intrinsic dimension by “linear regression on R”. This last step is not clear. What is being regressed? It is a regression between R and what?

5. In the equation in line 229, the variable “d” is not defined. Could you describe that equation?

6. Figure 3 is hard to see because the dots are very small (even when zooming into the figure). The authors conclude from that analysis that “we hypothesize that multi-modal visual models may initially interpret text as a textural feature in the earlier layers”. In my opinion, I do not think one can conclude anything about how text is encoded there by just looking at the result from figure 3. The text is small in the image. The representation in the first layers is likely to be dominated by image features that occupy large image regions.
But isn’t it better to interpret the result as if that representation in early layers is dominated by all the image features (not just text)? Clearly, the last layer can focus on smaller image regions that contain important information, and it separates all the information (image and text) and t-sne can differentiate among the 6 sets.

7. In figure 3, what happens with the nonsense text?

8. Could you describe the notation used in table 1? What does the number in Cons_80, … Irr_* means? I assume it refers to the font size as shown in the appendix, but I think it will be useful to point the reader to the appendix or to include a short description in the text somewhere in the lines 315-320 or in the table caption.

9. Once the reader arrives to section 5, there seems to be no connections between the experiments performed in section 5 and the analysis in the previous sections. The previous sections seem to be used only to support the observation that “early layers of visual models primarily rely on texture features rather than true semantic understanding”. But one could arrive to the same conclusion just from the experiments of section 5.

---

> ### Author Response · Authors · 2024-11-22
>
> We sincerely thank the reviewer for their valuable and detailed suggestions. Below, we provide responses that aim to clarify the concerns and enhance the presentation of our work. Key revisions are highlighted in blue in the PDF.
>
> - Q1. I would recommend making some questions softer. For instance, the question “do these models genuinely understand the semantics of the text or are they merely recognizing it as a visual pattern?” is a really difficult question.
>
> 	A1. We appreciate your thoughtful suggestion and fully acknowledge that the question of whether these models truly comprehend the semantics of text or simply recognize it as a visual pattern is highly complex. This question serves as the key motivation for our work, but we agree that a conclusive answer requires deeper and more extensive experimentation. In the revised version, we will soften some of the assumptions and conclusions, particularly around "genuine semantic comprehension," and will explicitly distinguish between our motivations, the specific problems we address, and the observations versus conclusions we draw.
>
> 	We see this question as an important direction for further exploration. Our work provides a preliminary investigation, focusing on empirical observations that highlight certain patterns and behaviors in visual models. While our findings are not definitive, we hope they offer useful insights and a foundation for future research in related areas.
>
>
> - Q2. Line 68, the sentence “Our findings reveal a non-linear pattern in representation” is repeated twice.
>
> 	A2. Thank you for pointing out the repetition. We will correct this in the revised version and ensure the manuscript is thoroughly proofread to eliminate any typos.
>
> - Q3. Maybe you could rename the orthographic pairs as “Paronyms”.
>
> 	A3. Thank you for your detailed suggestion. "Paronyms" indeed offers a more precise and comprehensive term compared to "orthographic." It better reflects the criteria and approach used in constructing our dataset. In the revised version, we will replace "Semantic Orthographic Pairs" with "Synonyms Paronyms Pairs" to improve the clarity of our presentation.
>
> - Q4.  In the algorithm 1, What is being regressed？
>
> 	A4. The goal of the regression step is to estimate the intrinsic dimension (ID) by fitting the distance ratios $\( R[i] \)$. These ratios, derived from the first and second nearest neighbors, are expected to follow a Pareto distribution. The likelihood of these ratios given the intrinsic dimension $\( d \)$ is expressed by the function $\( P(\mathbf{R} | d) \)$, which we maximize using linear regression.
>
> 	The regression does not simply fit the ratios directly but aims to maximize the likelihood function to estimate the intrinsic dimension that best represents the local geometry of the data. In the revised manuscript, we clarify this process and the connection to the Pareto distribution to ensure that this important detail is more clearly explained.
>
> - Q5. In the equation in line 229, the variable “d” is not defined.
>
> 	A5. The variable "d" represents the intrinsic dimension (ID). We clarify this in the explanation of the equation in the revised version.
>
> - Q6.1. Figure 3 is hard to see because the dots are very small.
>
> 	A6.1. We increase the size of the dots and add additional nonsense data. The updated figure replaces the original Figure 3 in the revised version.
>
> - Q6.2. I do not think one can conclude anything about how text is encoded there by just looking at the result from Figure 3.
>
> 	A6.2. We agree with the reviewer that the observation from Figure 3 should be interpreted as a hypothesis rather than a conclusive result. It serves as the foundation for our subsequent experiments and inferences. As discussed in our response to Question 1, the purpose of this content is to provide a perspective and interpretation on the matter, rather than offering definitive conclusions at this stage.
>
> 	We agree that early layers are generally influenced by the overall image features (not only text or objects). However, we do not think this is primarily related to text/object size. And our hypothesis is reasonable, since early layers typically focus on fine-grained details, while later layers capture more abstract, global features. Therefore, the results in the final layer possibly reflect a broader understanding of the image, supporting the separation of objects and text, as shown in our t-SNE results.
>
> 	To address the concern, we revise the presentation of our results to emphasize that the findings should be viewed as an initial observation rather than a conclusive interpretation. Hope this revision will enhance the rigor of our discussion and leave space for further exploration and refinement in future research.

---

> > ### Author Response · Authors · 2024-11-22
> >
> > - Q7. In figure 3, what happens with the nonsense text?
> >
> > 	A7. The nonsense text samples have been included in the updated Figure 3. We observe no significant differences in the t-SNE visualization results between the nonsense text and other overlaid text.
> >
> > - Q8. Could you describe the notation used in table 1?
> >
> > 	A8. We recognize that the notation may be unclear. In the revised version, we explicitly state that the numbers correspond to the font size.
> >
> > - Q9. The paper seems to have parts that are not well connected: the results on the intrinsic dimension (ID) seem disconnected from the defenses and results presented in section 5.
> >
> > 	A9. Thank you for pointing this out. We expand the explanation at the beginning of Section 5 to clearly connect the analysis in Section 4 with the proposed defense approaches, providing a smoother transition and improving the paper's coherence.

---

### Official Review · Reviewer_P9S9 · 2024-11-04

**Soundness:** 4
**Presentation:** 3
**Contribution:** 3
**Rating:** 6
**Confidence:** 5

**Summary:**

This paper investigates how vision-language models, particularly CLIP, process text in images, questioning whether they truly understand semantics or merely recognize visual patterns. Using a novel dataset and Intrinsic Dimension analysis, the authors find that texture heavily influences representations, even in later layers, with semantic understanding primarily emerging in the final block. They propose a defense against typographic attacks by fine-tuning this final block.

**Strengths:**

1. The ToT dataset, particularly the subset designed to disentangle orthography and semantics, is a valuable contribution and allows for a more nuanced investigation of how visual models process text.
2. The use of Intrinsic Dimension (ID) provides a quantitative measure of representational complexity, offering insights beyond qualitative visualizations. The analysis reveals a complex interplay between texture and semantics across different layers.
3. The proposed defense strategy of fine-tuning only the final block is a practical and potentially efficient approach, grounded in their analysis of representational changes across layers.

**Weaknesses:**

1. The analysis primarily focuses on CLIP. While CLIP is a representative vision-language model, exploring other architectures would strengthen the generalizability of the findings.
2.  While the proposed defense strategy shows promise, a comparison with existing defense mechanisms against typographic attacks is missing. This would provide a better context for evaluating the effectiveness of their approach.
3. While the paper analyzes the impact of text size and semantics, a more comprehensive ablation study is needed. For instance, exploring different font styles, text placements, and background complexities would further elucidate the interplay of texture and semantics.

**Questions:**

1. What are the long-term effects of fine-tuning only the final block on the model's performance over time? Are there any observed degradations in performance on non-typographic tasks?
2. The paper uses ImageNet-1k as the basis for the ToT dataset. How might the findings change if the dataset were based on a different image dataset with more diverse scenes and text occurrences?
3. The authors mention that "genuine semantic comprehension only emerges in the final block." Could you provide further evidence or analysis to support this claim? How do you define and measure "genuine semantic comprehension" in this context? How does this relate to the observed decrease in ID for consistent text overlays in the final block?

---

> ### Author Response · Authors · 2024-11-22
>
> We sincerely thank the reviewer for their thoughtful feedback. Below, we provide detailed responses to the concerns raised regarding our approach and results. The primary revisions are highlighted in blue in the PDF. We hope the clarifications and additional experiments effectively address your concerns.
>
> - W1. Exploring other architectures except for clip.
>
> 	A1. Thank you for raising this important question. To address it, we extend our analysis to include ShareGPT-4v, a generative multimodal model that differs significantly from CLIP’s discriminative architecture. ShareGPT-4v is designed to handle more complex and diverse generative tasks. The intrinsic dimension (ID) results for ShareGPT-4v are presented in Figure 9 of the appendix. Notably, we observe substantial ID differences in its final block, similar to CLIP, which reinforces the consistency of our findings across different model architectures.
>
> - W2. Including a comparison with existing defense mechanisms.
>
> 	A2. The comparison with existing defense methods is presented in Section 5.2. Table 4 (in the manuscript) shows performance against the standard typographic attack, while Table 5 evaluates defenses against two more challenging typographic attack scenarios that we propose. In Section 5.2.1, we make these comparisons more explicit to enhance clarity.
>
> - W3. Exploring different font styles, text placements, and background complexities.
>
> 	A3. The ToT dataset includes a wide range of font styles and colors, as detailed in the appendix (Font section), with text randomly positioned on the images. We will revise the appendix to provide additional details and ensure these settings are more clearly highlighted. Additionally, we have added experiments based on the Caltech101 dataset, which features different backgrounds compared to our ToT dataset. The results are shown in Figure 11 (in the appendix).
>
> - Q1. What are the long-term effects on non-typographic task performance over time?
>
> 	A1. This is a valid concern. We fine-tune the model for 5 epochs with a learning rate of 1e-4. The results, shown in Figure 10 (in the Appendix), indicate that accuracy improves slightly in the second epoch compared to the first epoch (reported in the main paper). Afterward, performance on non-typographic tasks gradually decreases, but eventually stabilizes near the original CLIP performance. While this slight decline is noticeable, the improvement in typographic attack defense is much more substantial, which we believe justifies the trade-off. In fact, the defense improvements far outweigh the minor performance loss on non-typographic tasks. Additionally, by selecting an optimal epoch based on validation performance (e.g., epoch 1-3), we can achieve improvements in both adversarial and original task performance.
>
> - Q2. How might the findings change if the dataset were based on a different image dataset with more diverse scenes and text occurrences?
>
> 	A2: We carefully considered diversity and representativeness when constructing the ToT dataset, which only includes categories corresponding to common real-world entities. To further validate our approach, we also conduct experiments using the Caltech101 dataset. The results, shown in Figure 11 (in the appendix) and following Table, demonstrate that our method remains effective, though slightly less so compared to ImageNet. We attribute this to the simpler backgrounds and lower resolution of the Caltech101 images, whereas ImageNet images have higher resolution and more complex scenes, making them more reflective of real-world data.
>
> | **Model** | **Disentangle** | **PAINT** | **Prefix** | **Avg** |
> |-----------|-----------------|-----------|------------|---------|
> | CLIP      | 43.3            | 50.0      | 47.2       | 46.8    |
> | Ours      | 72.1            | 52.7      | 52.2       | 59.0    |

---

> > ### Author Response · Authors · 2024-11-22
> >
> > - Q3. How do you define and measure "genuine semantic comprehension" in this context? How does this relate to the observed decrease in ID for consistent text overlays in the final block?
> >
> > 	A3. In this context, "genuine semantic comprehension" refers to the encoding complexity that is significantly influenced by semantic meaning, rather than by visual features or superficial textural details (which are, to some extent, unavoidable). This concept captures the extent to which the semantic meaning of the overlaid text shapes the model's encoding without being dominated by visual attributes.
> >
> > 	Regarding the observed decrease in intrinsic dimension (ID) for consistent text overlays in the final block, we hypothesize that when the text is semantically aligned with the image content, it reduces the overall semantic encoding complexity, leading to a decrease in ID. This reduction is most evident in the final block's representation, suggesting that the model achieves its most significant semantic comprehension, i.e., encoding complexity strongly driven by semantic content, in this layer.
> >
> > 	 further clarify, we believe that redefining this concept as "encoding complexity significantly related to semantic meaning" may better express its relationship with intrinsic dimension and highlight our intent more clearly.

---

> > > ### Author Response · Authors · 2024-11-26
> > >
> > > Dear Reviewer,
> > >
> > > Thank you for your thoughtful feedback and the time you have devoted to reviewing our work. As the rebuttal period is coming to an end, we hope that the revisions have effectively addressed your concerns. If there’s anything further you’d like to discuss, we’d be happy to engage.

---

### Meta-Review · Area_Chair_h7zq · 2024-12-20

**Metareview:**

This paper explores the extent to which the CLIP model truly comprehends textual semantics in images versus merely recognizing visual features. It challenges the prevailing notion that such models understand text, and to aid this investigation, the authors present a new dataset called ToT (Texture or Textual) that differentiates between the visual aspects of words and their meanings. Via an analysis using the Intrinsic Dimension they demonstrate that in the initial layers texture and semantics are in competition, with semantic comprehension largely emerging only in the final layer. The paper also proposes strategies to defend against typographic attacks through refinement of the final block.

The reviewers articulated a number of strong points in the paper. In particular, reviewers were generally unanimous in their appreciation of the  ToT dataset, especially its focus on separating orthographic features from semantics. Some reviewers also comment that utilizing Intrinsic Dimension is an intriguing approach for assessment of representational complexity and that it provides deeper insights into the relationship between texture and semantics across layers. The proposed strategy for defending against typographic attacks, which only involves fine-tuning only the final block, reviewers also found straightforward yet effective (as demonstrated by the analysis of representation shifts within the model and performance compared to the baseline).

However, the paper also has a number of weak points that outweigh its positive aspects:
- **Concentration on CLIP**: The analysis in the paper is predominantly centered on the CLIP model, and examining additional architectures would enhance the broader applicability of the conclusions. As one reviewer points out, the use of "how *visual models* understand texts in images" in the title leads the reader to assume the analyses and conclusions would be more general.

- **Limitations in Experimental Evaluation**: While the paper considers the effects of text and visual semantics, a more thorough ablation study is necessary, one which incorporates systematic typographical variations in order to better illustrate the relationship between texture and semantics.

- **Clarity**: Some sections of the paper seem disjointed, particularly the discussion of Intrinsic Dimension (ID) and the defense strategies in Section 5 without explicit an explicit link or clear motivation of the need for ID. Additionally, the exploration of ID for this specific task could be more detailed. The authors should clarify the rationale for incorporating ID and its relevance to the paper's objectives.

The general consensus that emerges is that this paper has some very interesting ideas in it, however it is in need of extensive revision in terms of clarity and motivation before it can be considered for acceptance.

**Additional Comments On Reviewer Discussion:**

The authors furnished a number of clarifications in rebuttal, which addressed some of the reviewer concerns. However, lingering questions regarding clarity of technical presentation and motivations for using the Intrinsic Dimension led to little-to-no enthusiastic support for accepting the paper.

---

### Decision · Program_Chairs · 2025-01-22

Reject